# *Burkholderia cepacia* Enhanced Electrokinetic-Permeable Reaction Barrier for the Remediation of Lead Contaminated Soils

**Yun He [†], Linlin Yang [†], Chiquan He \* and Feifei Wang \*** 

School of Environmental and Chemical Engineering, Shanghai University, Shanghai 200444, China
\* Correspondence: cqhe@shu.edu.cn (C.H.); feifeiwang@shu.edu.cn (F.W.)
† These authors contributed equally to this work.

**Abstract:** The combination of electrokinetic (EK) and permeable reactive barrier (PRB) is a potentially effective technology for the remediation of heavy-metal-contaminated soils, but high energy expenditure limits its application in practice. In order to further improve the remediation efficiency and reduce the cost, some improvements were made in this study in terms of new PRB material, the spatial configuration of the rod electrode, and the microbial enhancement. Differently from previous powder PRB materials, six thin-film PRB materials were prepared using cheap natural attapulgite (ATP) and metal salts. PRB is a tough material that can be plugged and pulled out in engineering. The heavy metals adsorbed on it can be extracted from the soil, eliminating the risk of subsequent secondary pollution. Therefore, it has a strong operational ability. Among them, the FeMn-ATP material exhibited the best adsorption performance ($2521 \pm 377.1$ mg/kg) for Pb. The results of the transmission electron microscope, X-ray diffractometer, and Fourier-transform infrared spectroscopy showed that iron and manganese were successfully loaded on the material. The *Burkholderia cepacia* pre-treatment led to soil pH decrease and the dissolution of Pb, and the morphological composition of Pb in the soil was also changed. In the microbial group, the adsorption amount of Pb by PRB at the anode and cathode increased by 69.1% and 42.1%, respectively. The concentration of the residual lead in the anode soil was significantly lower than that in the control group without microorganisms, and the removal rate of Pb was increased by 26%.

**Keywords:** bioaugmentation; hexagonal electric field; heavy metal; PRB; sheet

## 1. Introduction

A staggering amount of lead (Pb) is discharged into the biosphere every year as a result of human activities. Pb is highly toxic and difficult to degrade. In addition, its direct or indirect absorption and subsequent accumulation in human tissues will cause serious health problems [1,2]. The China Soil Pollution Survey published in 2014 indicated that 16.1% of arable land is polluted, and inorganic pollutants, particularly cadmium (Cd), is found in high quantities in 7% of the surveyed sites [3]. It is of great significance to study the remediation of soils contaminated by heavy metals. The remediation of Pb-polluted soils can be achieved through physical and chemical technologies [4] and through bioremediation [5]. According to the actual engineering implementation conditions, the current soil remediation technology can be divided into two categories, in situ remediation and ectopic remediation. Wan et al. reported the effectiveness of Pd/Fe PRB installed in a hexachlorobenzene (HCB)-contaminated soil coupled with EK technology. The results showed that the removal rate of HCB increased by four times compared to EK alone [6]. It can be seen that EK-PRB remediation technology has great advantages and application potential.

As stated in our previous study, the combination of EK and PRB possessed a synergistic effect for heavy metal removal [7]. Many studies focused on the enhancement methods of EK remediation and the modification of PRB materials [8]. At present, there are several



types of PRB materials widely used. The first are adsorbents, such as active aluminum, activated carbon, coal, natural zeolites, peat, etc., which take advantage of their loose porosity and large surface area to adsorb pollutants to achieve the purpose of cleaning up pollution and purifying soil. Mahabadi et al. have shown that natural zeolite reduced the leaching of Cd in contaminated soil [9]. Natural zeolite is difficult to collect from soil in practice because of its loose structure. At present, the research on the pollution control of nanomaterials is mainly focused on sewage treatment, while the research on soil remediation is limited [10]. The second type is precipitant [11–13], such as lime, limestone, nitrate, ferrous salt, etc. The solubility of metal ions is relatively low so they easily precipitate. At the same time, acid ions and heavy metal ions are introduced to separate heavy metal elements from the soil through coprecipitation. The third category is reductant, which mainly uses zero-valent metals, such as zero-valent iron (Fe0) and zero-valent copper (Cu0). The zero-valent Fe/Cu nanoparticles in contaminated soil had a reduction efficiency for Cr (VI) c of more than 99% in weakly acidic conditions, and the price of this kind of PRB is high, which requires the morphology of contaminated soil and the valence of heavy metals, and its practical applications are limited [14]. Attapulgite is a kind of water-rich magnesium-rich aluminosilicate clay mineral with a chain-like structure. The unique layer–chain crystal structure and very fine rod-like and fibrous crystal morphology cause it to have a high specific surface area and good adsorption properties. The modified attapulgite by acid, alkali, and heat treatment can greatly improve its adsorption properties [15]. Considering the economy, convenience, and reusability of a PRB material during the remediation process, the authors proposed to apply modified sheet PRB material based on attapulgite to replace the powder material in this study.

Electrokinetic remediation refers to the application of direct current at both ends of the contaminated soil to form a voltage gradient, and the pollutants in the soil are transferred to both ends of the electrode by electrodialysis, electromigration, and electrophoresis under the action of an electric field, so as to achieve the technology of remediation contaminated soil. Compared with the uniform electrodynamic system of the plate electrode, the current intensity of the non-uniform electrodynamic system of the regular hexagonal rod electrode matrix is larger, the range and degree of influence on soil moisture are smaller, and the operating stability of the system is higher. Kim et al. reported that the hexagonal rod electrode device increases the effective area of the electric field and can remove heavy metal pollution more effectively [16]. Liu used a hexagonal rod electrode matrix to repair Cd and Pb in artificially polluted kaolin [17].

Microorganisms play an important role in the repair of heavy metals. First of all, under heavy metal stress, microorganisms can produce a large number of extracellular secretions, which can provide a large number of anionic groups to combine with heavy metals to form precipitates through complexation, chelation, and other ways. In order to reduce the biological toxicity of heavy metals, Park et al. applied phosphate-solubilizing bacteria and phosphate rock powder to conduct passivation remediation in lead-contaminated soils [18]. Li et al. found that the combination of phosphate-solubilizing fungi and fluoroapatite could effectively solidify lead in soil [19]. Secondly, in the aspect of biosorption and enrichment, microbial extracellular polymers can combine with heavy metal ions so that heavy metal ions in the environment are adsorbed [20–22]. Amoroso et al. found that Mycorrhizal fungi and humus-decomposing bacteria isolated from the soil near a mercury mine can enrich Hg from the soil [23]. Field et al. found that Rhizopus can rapidly adsorb a variety of heavy metal ions, such as $Ni^{2+}$, $Pb^{2+}$, and $Cd^{2+}$. In addition, they also play an important role in biotransformation. The redox of microorganisms to heavy metal elements is one of the important detoxification mechanisms [24]. Microorganisms can change the valence state of the presence of heavy metals by autocrine enzymes, reducing the valence state from highly toxic to that of low toxicity [25]. For example, Macaskie et al. [26] isolated a bacterium that can produce citrate phosphatase and break down organic 2-phosphoglycerol to produce $HPO_4^{2-}$, $HPO_4^{2-}$ precipitates with $Cd^{2+}$ to form $CdHPO_4$, and therefore alter the ecological toxicity of Cd. Qudsia et al. demonstrated that the release of organic acid by

microorganisms to solubilize inorganic P bound to soil colloids is an important mechanism in which $COO^-$ (carboxyl group) and $OH^-$ (hydroxyl ion) act as chelators of cations such as Fe, $Al^{3+}$, and $Ca^{2+}$ and compete for P adsorption sites in soil [5]. Naveed et al. found the ameliorative role of applied biochar and seed inoculation with B. phytofirmans PsJN through improved growth, physiological, biochemical, and antioxidative defense responses in mung beans grown under normal and Pb-spiked soils [6]. However, there are few studies on the synergistic effect of the combination of microorganisms and EK-PRB technology on remediating lead-contaminated soil under the hexagonal electric field.

The objectives of this study were to (1) prepare shaped PRB materials with low cost and easy operation in practice and choose the optimal one for Pb adsorption, (2) investigate the adsorption kinetics of Pb by the optimal PRB material, (3) evaluate the enhancement effect of microbial pre-treatment on Pb removal by EK-PRB technology, and (4) analyze the synergistic mechanism of dominant microorganisms and electrokinetic remediation. Overall, this study aimed to provide potentially efficient technologies for Pb-contaminated soil remediation.

## 2. Materials and Methods

### 2.1. Chemicals and Soils

Sinopharm Chemical Reagent Co., Ltd. (Shanghai, China) supplied Sodium alginate, Polyvinyl alcohol, Ferric trichloride, Potassium permanganate, Hydrogen peroxide, Calcium chloride, Sodium hydroxide, Lead nitrate, Hydrogen chloride, Nitric acid, and Hydrofluoric acid. Attapulgite (ATP) was purchased from Mingmei Co., Ltd. (Hefei, China). All of the chemicals used in the study were of at least analytical grade.

The soil in this experiment was taken from the original site of a film machinery factory in Shanghai. The buildings on the contaminated land have been demolished, and the soil was collected at a depth of 20–50 cm. In order to better simulate the actual situation, the actual contaminated soil was naturally air-dried, crushed in the grinder, and then sifted through the 2-mesh sieve. The physical and chemical properties of the tested soil are shown in Table 1.

**Table 1.** Physical and chemical properties of the soil.

| Parameters | Value |
|---|---|
| pH | 8.41 |
| Organic matter (g/kg) | 26.0 |
| Total salt content (g/kg) | 1.08 |
| Sand grain 2–0.05 (mm) | 40% |
| Powder granule 0.05–0.002 (mm) | 59% |
| Clay particle <0.002 (mm) | 1% |
| Cation exchange capacity (cmol( + )/kg) | 8.0 |
| Bicarbonate radical (g/kg) | 0.52 |
| Pb (mg/kg) | 136 |
| Soil type | Sandy loam |
| Appearance | Grey-yellow fluvo-aquic soil |

The bacteria used in this study was *Burkholderia cepacian*, which was purchased from the China General microbial strain Preservation and Management Center. The bacteria were cultured in 98.2% nutritious gravy medium containing 1% tryptone, 0.3% yeast extract and 0.5% NaCl. After shaking at 30 °C for 12 h, the microbial cells were collected by 3000 rpm centrifugation and 10 min. The bacteria were washed with aseptic water twice, and the OD 600 nm value was adjusted to 35.25.

### 2.2. Experimental Processes

### 2.2.1. Preparation of Sheet PRB Materials

In order to improve the utilization and recycling of PRB, sheet PRB material was proposed to be used in conjunction with EK for the first time. The PRB material was Sodium alginate (SA)/$FeCl_3$/$MgSO_4$/$KMnO_4$/polyvinyl alcohol (PVA)/ATPcomposite hydrogel. Firstly, ATP was modified by precipitation and calcination to obtain the Fe-ATP, Mg-ATP, FeMg-ATP, FeMn-ATP, and FeMnMg-ATP composites, respectively. Then, 0.25 g of each of the six kinds of ATP were dispersed in 25 mL of deionized water to form a clay suspension under intense stirring. Then, 0.5 g of PVA was added to the suspension and heated in a water bath at 95 °C. After the complete dissolution of PVA, 0.5 g of SA was added to form a homogeneous mixture by stirring. The mixture was injected into a square mold with the size of $10 \times 10 \times 1$ cm$^3$ using a syringe, followed by a freezing treatment in a refrigerator at $-20$ °C for 20 h and thawing at room temperature for 4 h. After three cycles of freezing and thawing, it was placed in a freeze dryer for freeze–drying. Subsequently, it was immersed in 100 mL of $CaCl_2$ solution (5 wt%) for 4 h in order to crosslink completely with $Ca^{2+}$, and then taken out and rinsed three times with deionized water. The prepared sheet material was used as a PRB for subsequent experiments.

### 2.2.2. PRB Screening through Plate Electrode EK-PRB Experiment

Figure 1 shows a schematic diagram of the EK-PRB remediation reactor. In the polymethyl methacrylate experimental container (20 cm length $\times$ 10 cm width $\times$ 10 cm height), about 2 kg of contaminated soil was filled between the two electrodes. The plate graphite electrodes (10 cm length $\times$ 10 cm width $\times$ 1 cm height) were directly inserted into soils without electrolytic cells and connected to a direct current (DC) power supply through wires. The prepared PRB materials were placed next to the graphite electrode in the contaminated soil. The DC voltage was 20 V, and deionized water was added every 2 days to maintain a 30% water content.

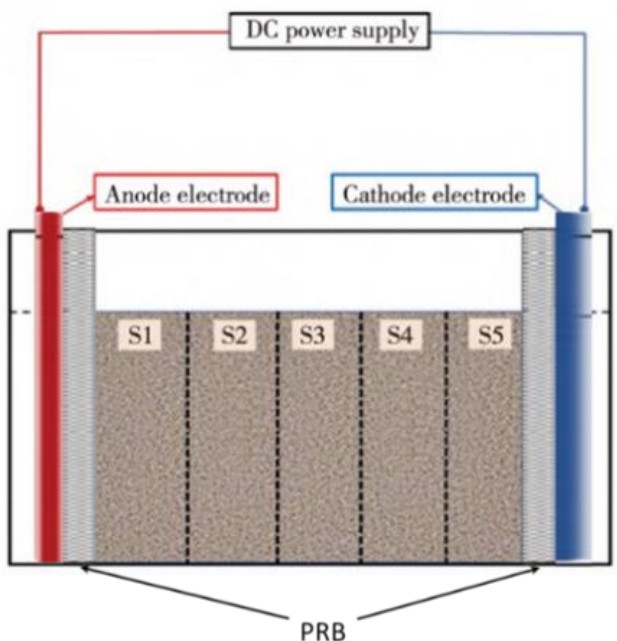

**Figure 1.** Schematic diagram of the sampling area of the plate electrode device.

### 2.2.3. Adsorption Kinetics of Heavy Metal Ions

Experimental Design of adsorption Kinetics of Pb: several pieces of FeMn-ATP sheet material weighing 2 g were placed into conical bottles containing $Pb^{2+}$, then stirred on a $140 \pm 2$ r/min shaker. Then, the 10 mL $Pb(NO_3)_2$ liquids were taken at 0 min, 10 min, 30 min, 1 h, 2 h, 4 h, and 6 h, respectively, then loaded into small test tubes. The concentrations of $Pb^{2+}$ were measured by inductively coupled plasma emission spectrometer (Avio 200 ICP-OES, Perkin Elmer).

Experimental Design of adsorption Thermodynamics of Pb: the solutions of $Pb^{2+}$ ion concentrations of 10, 20, 50, 100, 150, 200, 300, 400, 500, 600, 800, and 1000 mg/L were prepared. Then, 100 mL was added into a group of 200 mL beakers, then FeMn-ATP permeable reaction barrier was added, static adsorption for 16 h, and sampling for 10 mL. FeMn-ATP permeable reaction barrier at different initial concentrations of $Pb^{2+}$ ions.

### 2.2.4. Characterization of Synthetic PRB Sheet Material

After preparation, PRB material characterization was carried out to determine their properties. The surface structure was observed using a biological transmission electron microscope (JEM-1400 Nippon Electronics Co., Ltd., Dongjing, Japan). The crystal phase analysis and composition structure of the samples were studied by X-ray diffractometer (18KW/D/MAX2550VB/PC Japan Neo Motor Co., Ltd., Dongjing, Japan). The functional groups and chemical bonds of the samples were measured using Fourier-transform infrared spectrometer (7800-350/CM).

### 2.2.5. Experiment of Rod Electrode Enhanced by Microorganism

Figure 2 shows a schematic diagram of the EK-PRB remediation set-up. The EK device was mainly composed of the polymethyl methacrylate experimental container [27]. An appropriate amount of actual contaminated soil was filtered with a sieve to remove the grass roots, stones, and other impurities and then placed into the experimental container. Deionized water was then added to the soil, and the mixture of soil and water was stirred. The soil thickness was 8.5 cm. Seven columnar graphite electrodes coated with PRB material (20 cm length and 2 cm diameter) were inserted in the central position according to the regular hexagonal layout. The regular hexagonal central electrode was the cathode, the six vertex electrodes were the anodes, and the distance between the heterosexual/homosexual electrodes was 35 cm. The voltage between the anode and cathode electrodes was 35 V. The experiment was divided into two groups: the control group without microorganism addition and the experimental group with microorganisms. After sterilizing the actual contaminated soil with a sterilizer at 121 °C for 3 consecutive days, *Burkholderia cepacia* solution was added to each group of experimental soil and stirred evenly, and an appropriate amount of deionized water was added, which keep the soil moisture content at 30% [7]. The repair lasted for 14 days. During the EK-PRB remediation, the soil samples were collected on day 0 (0 h, day 2 (48 h), day 4 (96 h), day 6 (144 h), day 8 (192 h), day 10 (240 h), day 12 (288 h), and day 14, from sites A1, A2, A3, and A4 on line A (anode–cathode connection line) and sites B1, B2, B3, and B4 on line B (the vertical line between the anode and anode), as shown in Figure 2. The soil samples were collected to monitor the heavy metal content and microbial community structure.

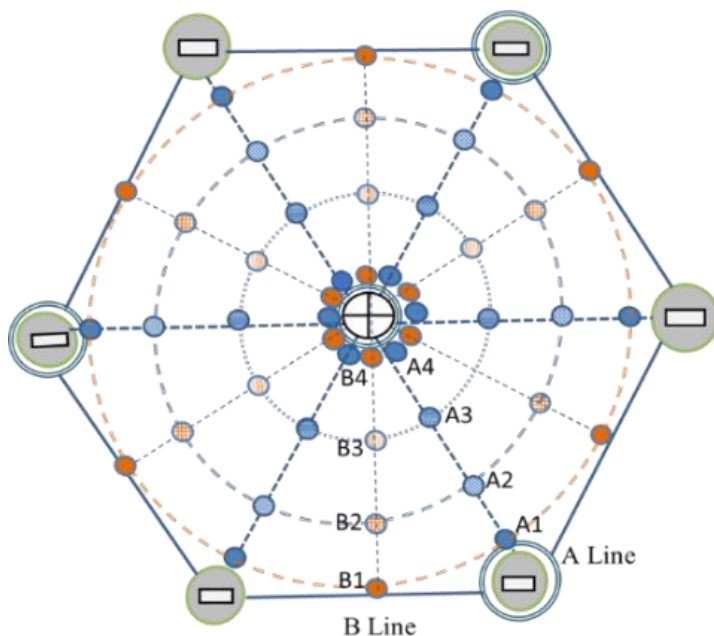

**Figure 2.** Soil sample collection sites and electrode position of EK remediation. Three cathode and one anode electrode with blue coil were wrapped by the optimal PRB material. The other cathode electrodes were not wrapped by PRB material.

*2.3. Analytical Methods*

A cylindrical glass tube with a diameter of 1 cm and a length of 10 cm was used to collect the soil samples for pH and HM analysis. An appropriate amount of the soil sample was used to analyze the moisture content, which was measured by weight loss after heating the soil at 105 °C for 8 h. The soil pH was quantified by a pH meter (PHS-25) in a suspension with a soil–water mass ratio of 1.0:2.5.

The XRD patterns of the samples were recorded using an X-ray diffraction (18KW/D/ MAX2550VB/ PC, Dongjing, Japan) with CuKα radiation. FTIR was used to detect the surface functional groups by an FTIR spectrophotometer (7800-350/CM, New York, NY, USA) using the KBr wafer technique.

The procedure for measuring the Pb content in the PRB material is as follows. Then, 0.5 g of the PRB wet sample material was placed into a 55 mL polytetrafluoroethylene digestion tube. Then, 6 mL of concentrated nitric acid, 2 mL of concentrated hydrochloric acid, and 1 mL of hydrofluoric acid were added in turn. The graphite electrothermal digestion instrument in the ventilation cabinet was heated at 120 °C for 4 h, which was repeated until PRB disappeared and the solution was clarified. The liquid was filtered into the 50 mL polyethylene tube using ultra-pure water. The concentrations braof $Pb^{2+}$ were measured by an inductively coupled plasma atomic emission spectrometer (Avio 200 ICP-OES, Perkin Elmer, Waltham, MA, USA).

The heavy metals of $Pb^{2+}$ in the soil samples were extracted by acid digestion. Specifically, $0.1 \pm 0.0003$ g of soil after drying and sieving was mixed with $HNO_3$ and $HClO_4$ (*v*/*v*, 4:1) for digestion at 165 °C for 4 h, and then HF and $HClO_4$ (*v*/*v*, 5:1) were added to extract $Pb^{2+}$ from the soil samples. After being filtered through a microfiltration membrane of 0.45 μm, the samples were quantified by inductively coupled plasma-atomic emission spectroscopy (Avio 200 ICP-OES, Perkin Elmer I).

To determine the binding forms of Pb (II) in the soil samples before and after EK-PRB remediation, the selective sequential extraction approach was performed by Tessier gradual separation technology [28].

SPSS software was used for statistical analysis in each treatment group. The significant differences among the different treatment groups were analyzed by single factor analysis

of variance (One-way ANOVA), and the significant differences among multiple groups were analyzed using the map-based HSA method. All of the graphics were completed in Origin 2016.

## 3. Results and Discussion

### 3.1. Optimization of PRB for Pb Adsorption

In order to choose the optimal PRB material for $Pb^{2+}$ adsorption, Figure 2 presents the adsorption performance of several PRB materials. It can be seen that the adsorption quantity of $Pb^{2+}$ in the PRB materials are ranked as FeMn-ATP > FeMg-ATP > Mg- ATP > FeMnMg-ATP > ATP > Fe-ATP (in Figure 3). The FeMn-ATP adsorbed the most $Pb^{2+}$, reaching $2521 \pm 377.1$ mg/kg. Therefore, the material of FeMn-ATP was preferred as the optimal PRB for adsorbing Pb. In addition to electrostatic adsorption, the adsorption mechanism of $Pb^{2+}$ by FeMn-ATP might be that $Pb^{2+}$ replaced the manganese ions loaded on ATP in the solution. The displacement reaction formula is shown in Equations (1) and (2) [29,30].

$$\text{Mn-OH} + Pb^{2+} \rightarrow \text{Mn-O-Pb}^+ + H^+ \tag{1}$$

$$2\text{Mn-OH} + Pb^2 \rightarrow (\text{Mn-O})_2\text{Pb} + 2H^+ \tag{2}$$

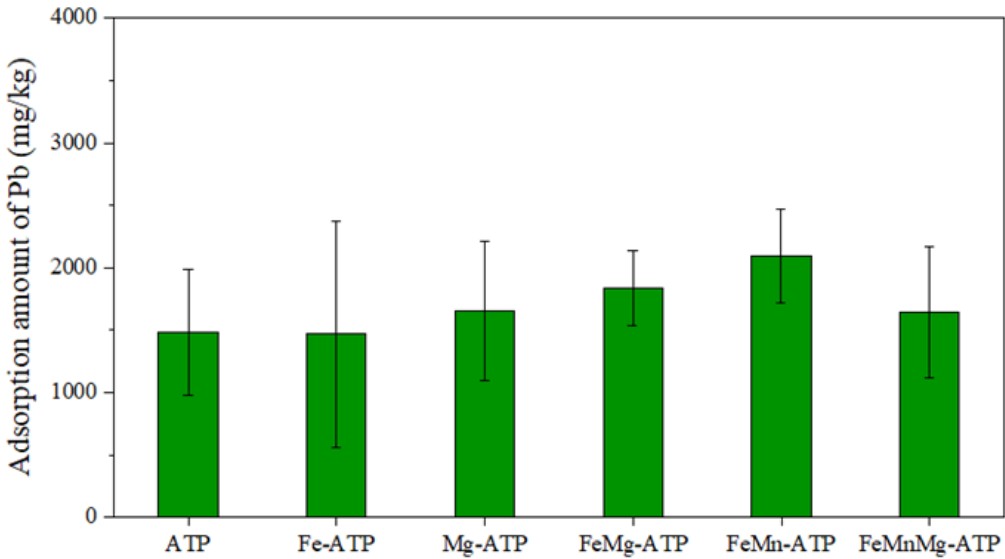

**Figure 3.** Adsorption performance of PRB materials on $Pb^{2+}$. Data are the mean of replications.

### 3.2. Adsorption Capacity of the Optimal PRB Material

It can be seen from Figure 4 that after 14 days of electrification, the adsorption capacity of FeMn-ATP for the lead at the anode is 136 mg/kg, and the adsorption capacity for the lead at the cathode is 52.3 mg/kg. The results showed that the $Pb^{2+}$ ions migrated to the permeable reaction wall and reacted with the fillers to precipitate, adsorb, and intercept, thus reducing the content of soil pollutants [31]. In the process of electrokinetic remediation, the pH value of the soil solution with a charge of zero is called the point of charge zero (ZPC). If the pH of the soil solution is higher than ZPC, it will contribute to the adsorption of cations. The pH increases gradually from the anode to the cathode, and the higher pH at the cathode is beneficial to the adsorption of heavy metal cationic pollutants. The content of lead in the anode of soil decreased obviously after 14 days. The obvious increase in the lead content in cathode after 14 days is due to the fact that $Pb^{2+}$ moves to two stages through electroosmosis and electromigration in the presence of an electric field and is absorbed by the ATP membrane of the poles, which can be explained by the obvious increase in the lead content in the ATP membrane. The lead content of anode ATP is obviously higher than that of the cathode, which may be due to the fact that under the action of an electric field, the

electrolysis of water on the electrode surface is shown in Equations (3) and (4), the anode $H^+$ increases, the anode pH decreases, the cathode increases, the plate and ATP are positively charged, the soil capillary inner surface is positively charged, and the solution is negatively charged, so the electroosmotic flow flows from the cathode to the anode in Figure 4b. At the same time, $Pb^{2+}$ is adsorbed by ATP at the cathode due to the electromigration of positive charge to the cathode. The amount of Pb through electromigration is less than that of electroosmotic migration, so the amount of ATP absorbed by anode ATP is obviously larger than that at the cathode.

$$\text{Anode reaction: } 2H_2O - 4e \rightarrow O_2\uparrow + 4H^+ \tag{3}$$

$$\text{Cathode reaction: } 2H_2O + 2e \rightarrow H_2\uparrow + 2OH^- \tag{4}$$

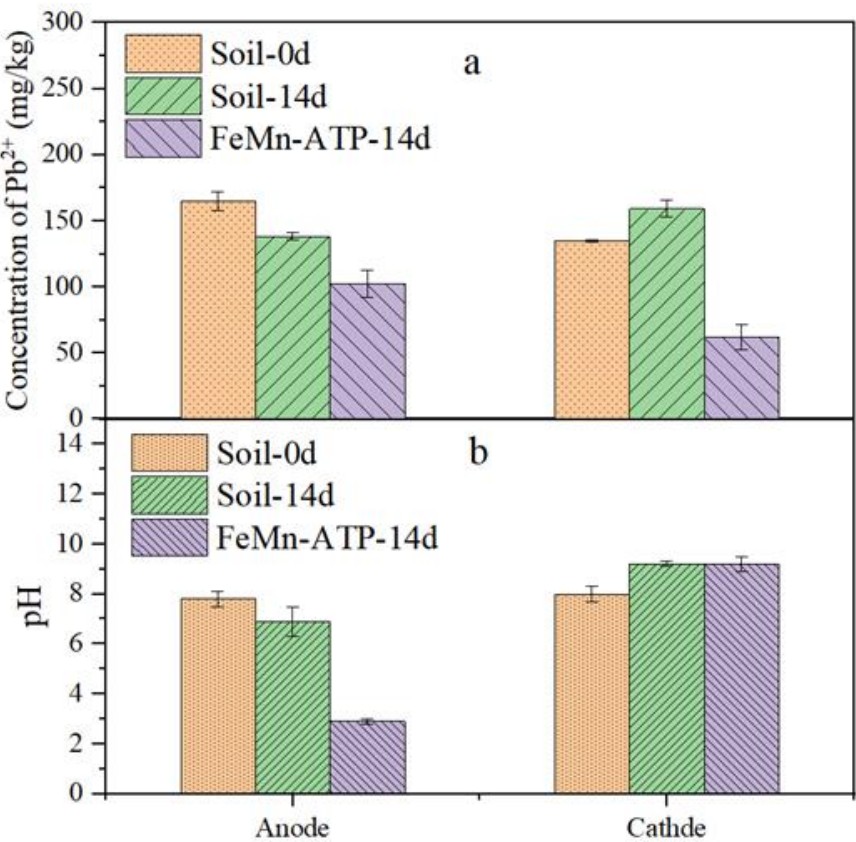

**Figure 4.** Adsorption capacity of FeMn-ATP (**a**) and change of pH (**b**) on actual contaminated soil in a plate-like electrode device.

During the adsorption experiment of the FeMn-ATP material, the change in the $Pb^{2+}$ concentration in the FeMn-ATP material with time is shown in Figure 5a. It can be observed that the adsorption process fits for quasi-first-order reaction kinetics with a correlation coefficient $R^2$ of 0.9520, while it cannot be fitted by the second-order kinetic well, indicating that the $Pb^{2+}$ adsorption by the FeMn-ATP material was mainly a physical adsorption.

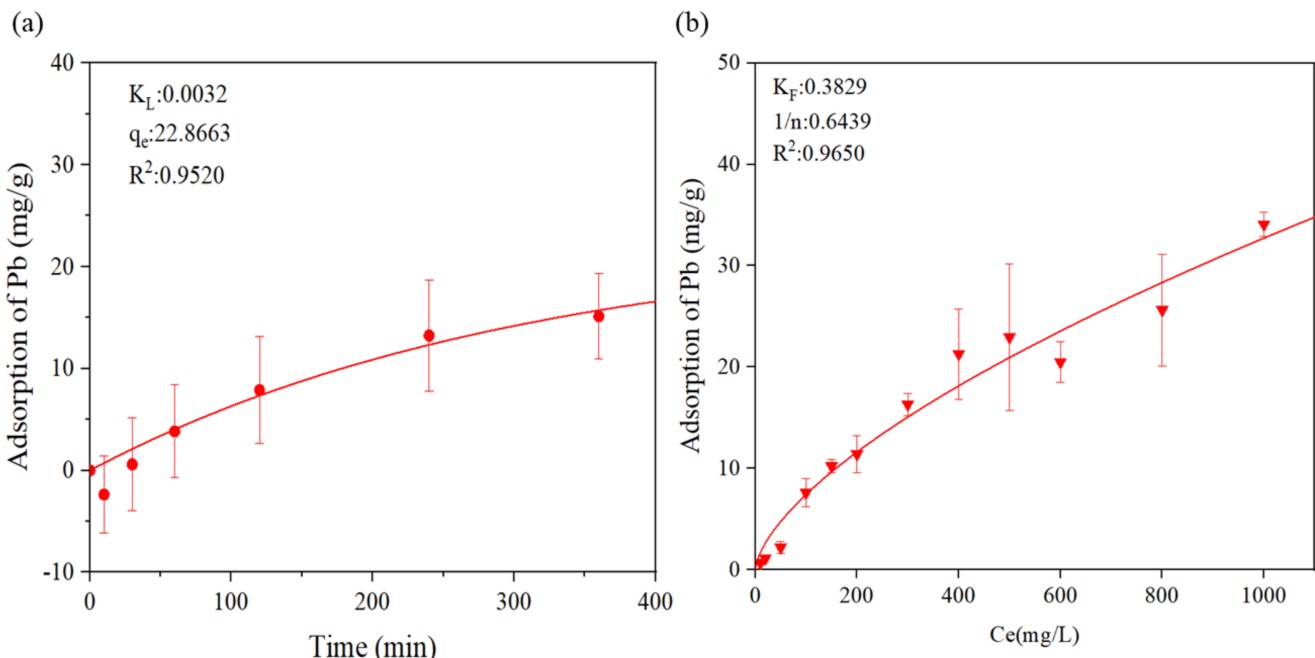

**Figure 5.** Adsorption kinetic curve (**a**) and adsorption isotherm (**b**) of $Pb^{2+}$ by FeMn-ATP material.

Figure 5b presents the adsorption isotherm, reflecting the adsorption capacity of the adsorbent at different equilibrium concentrations. $K_F$ is the Freundlich adsorption coefficient relating to the amount, property, and temperature of FeMn-ATP. $1/n$ is the Freundlich adsorption constant, which in ranges of 0.1–0.5 and >2 means easy adsorption and difficult adsorption, respectively [32]. In this experiment, $1/n$ of FeMn-ATP to $Pb^{2+}$ was 0.6439, closing to 0.5, indicating that $Pb^{2+}$ has a strong adsorption capacity. The adsorption isotherm fits for Freundlich model instead of Langmuir model. The fitness was good, with a correlation coefficient of 0.965, indicating that the adsorption mechanism may be heterogeneous multi-layer adsorption. In this paper, the adsorption effect of FeMn-ATP molded PRB material for lead is better than that of some granular or powdered PRB, with a lead saturation of 17.5131 mg/g reported by Xie et al. [33], so it has good prospects for engineering applications.

### 3.3. Characteristics of the Optimal PRB Material

The transmission electron microscope picture of the FeMn-ATP material is shown in Figure 6. It can be seen from Figure 5a that the original ATP is a slender rod-like structure, which is wound around each other to form a formed wheat-like structure. The oxide structure of ATP loaded with iron and manganese is shown in Figure 6b–d. Additionally, some non-uniform black particles can be clearly observed on the surface of ATP, indicating the successful loading of iron and manganese oxides by co-precipitation.

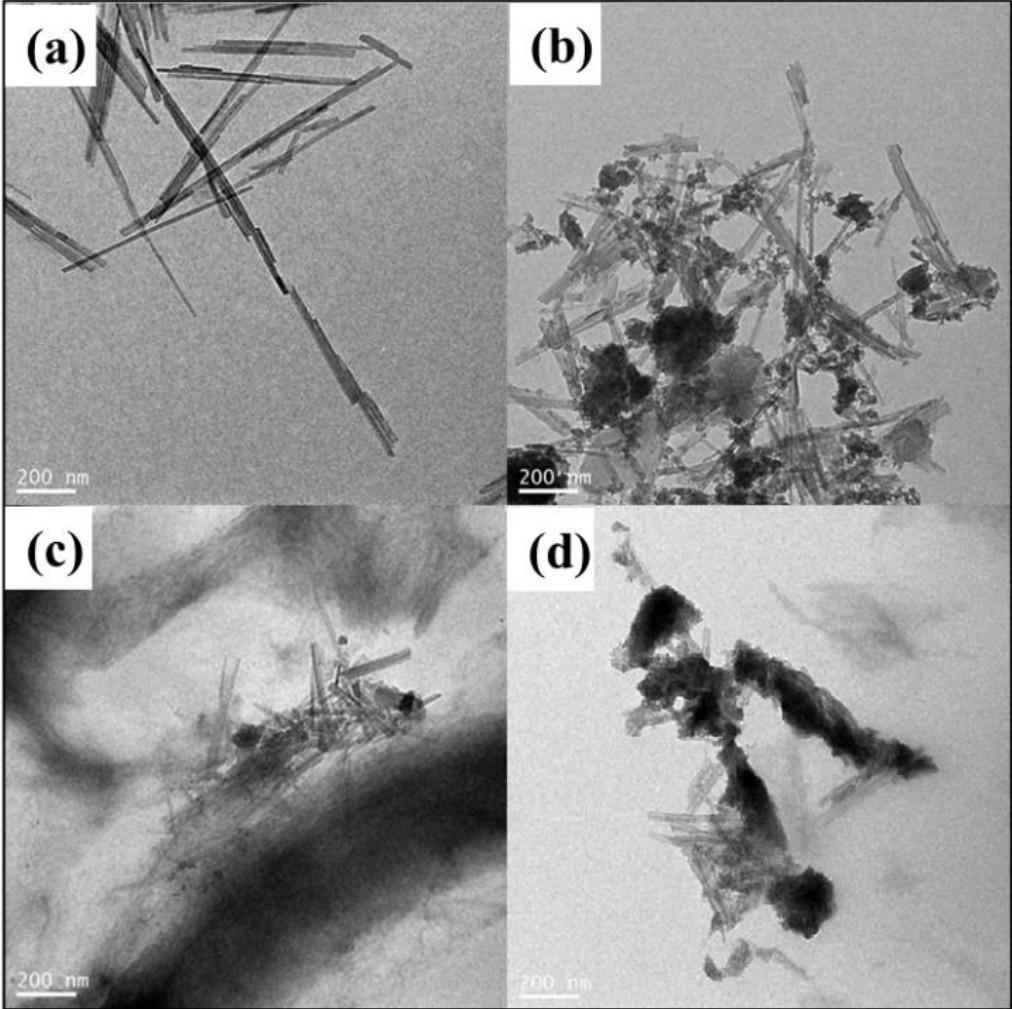

**Figure 6.** PRB transmission electron microscope diagram; (**a**) natural gravimetric clay, (**b**) FeMn-ATP powder, (**c**,**d**) FeMn-ATP permeable reaction barrier.

The XRD patterns of the ATP, FeMn-ATP powder, and FeMn-ATP permeable reaction wall are shown in Figure 7a. The results show that FeMn-ATP had not only the characteristic peaks of ATP but also the typical characteristic peaks of FeMn oxides at 32.7°, 43.0°, and 46.0°, indicating that Fe and Mn were successfully loaded onto the surface of ATP. The shaped FeMn-ATP sheet also had characteristic peaks of ATP and FeMn oxides at 24.5° and 48.1°, but the corresponding peaks were very small. The reason might be that sodium alginate with low crystallinity encapsulated ATP and FeMn oxides, resulting in a decrease in the crystallinity of the overall structure [34].

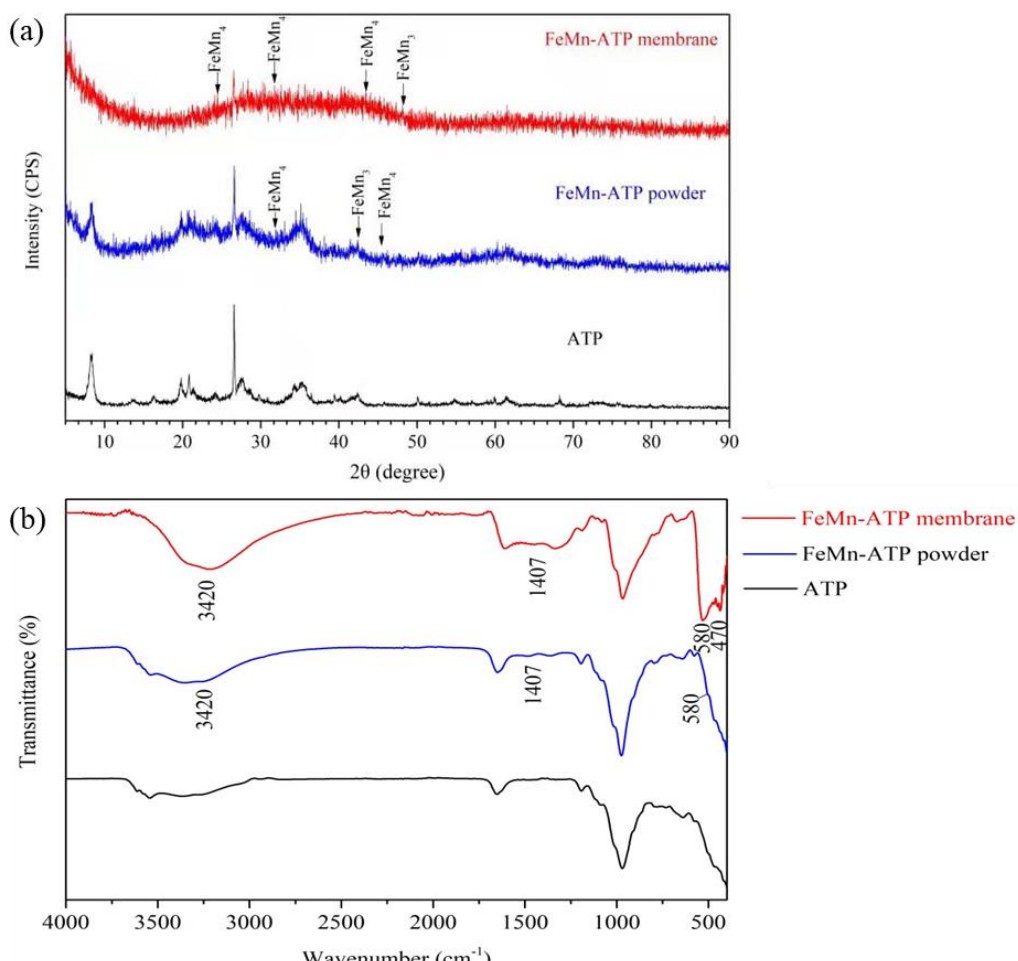

**Figure 7.** (**a**) X-ray diffraction pattern of Natural gravite, FeMn-ATP powder, FeMn-ATP permeable reaction barrier; (**b**) FTIR spectra of Natural graverite, FeMn-ATP powder, FeMn-ATP permeable reaction barrier.

The FTIR patterns of the ATP, FeMn-ATP powder, and FeMn-ATP sheet materials are shown in Figure 6b. In the FTIR spectrum of the pristine ATP, the characteristic peaks at 470, 580, 1407, and 3420 cm$^{-1}$ appeared. The peak at 470 cm$^{-1}$ corresponds to the stretching vibration of the Mn-O bond. The peak at 580 cm$^{-1}$ is attributed to the absorption peak of Fe-O. The characteristic bands at 1407 and 3420 cm$^{-1}$ are attributed to the asymmetric tensile vibration of $CO_3^{2-}$ in $MnCO_3$ and the molecular vibration of O-H of ATP [35], respectively. The appearance of the above characteristic peaks further indicates the successful preparation of FeMn-ATP material.

### 3.4. Changes of Electric Current and pH during Microorganism-EK-PRB Remediation

As can be seen from Figure 8, the current in the BC group with Brukholderia Cepacia pre-treatment gradually decreased from 146 mA to 31.5 mA, and it in the control group decreased from 122 mA to 14 mA during the remediation process. The decreasing trend of the current with the remediation time is in line with previous studies [7]. The current in the BC group decreased more slowly than that in the control group, and the average current in the BC group was higher than that in the control group, indicating that after the addition of *Burkholderia cepacia*, the number of mobile ions in the whole system was increased probably due to the production of organic acids and phosphorus solubilization by this microorganism. At the same time, the increase in the H$^+$ concentration acidified the soil, promoted the dissociation of soil heavy metal ions, and kept the current at a certain intensity during EK-PRB remediation.

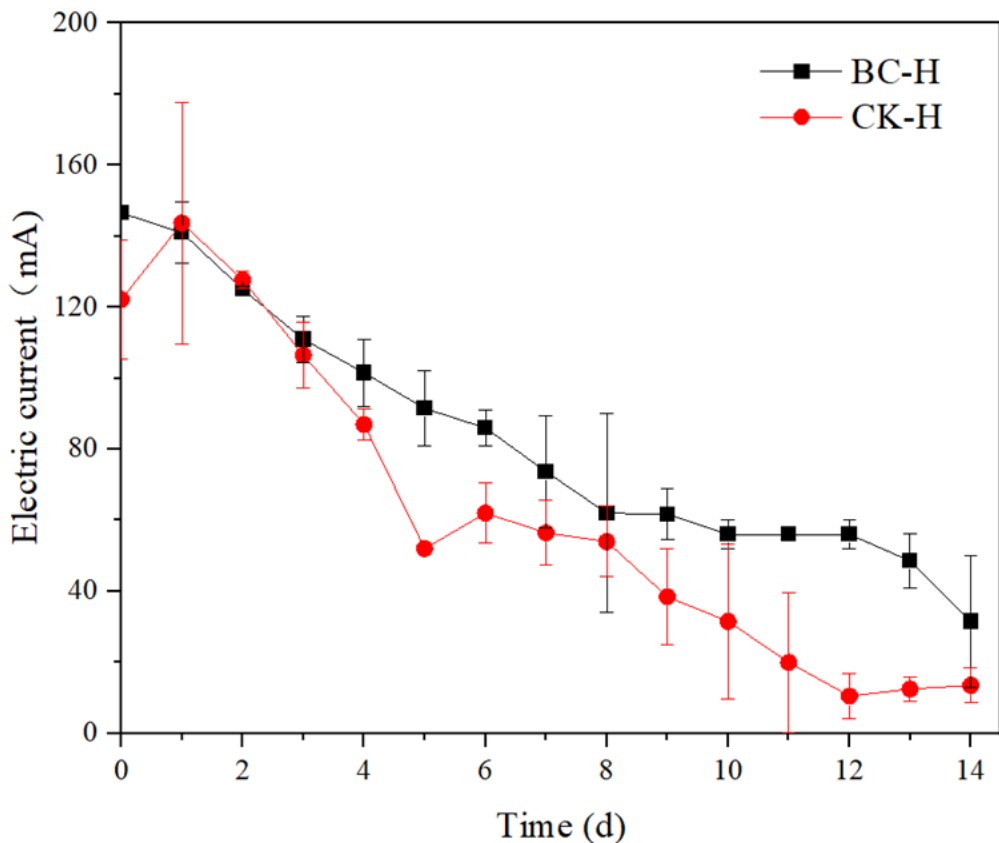

**Figure 8.** Current changes in the enhanced electrokinetic repair process of *Burkholderia cepacia*.

Soil pH is an important parameter affecting the transformation of heavy metals [36]. Figure 9 presents the current changes in the enhanced electrokinetic repair process of *Burkholderia cepacia* (the measured value drawn by interpolation method on EVS software). It can be seen from Figure 9a that the soil pH of the control group was unevenly distributed in the space after remediation. Most of the remediation areas were in the alkaline region, the anode soil was extremely acidic, and the cathode soil was extremely alkaline. Before the remediation, the initial pH was 8.5. After the remediation, in the control group, the pH of lines A and B changed greatly: the pH of points A1, A4, B1, and B4 became 4.0, 11.6, 9.2, and 11.6, respectively. It can be seen from Figure 9b that the soil pH in the BC group increased in concentric circles from anode to cathode and the soil pH near the cathode reached 10.2. After 14 days' remediation, the change range of pH on line A was larger than that on the B line. The pH of points A1, A4, B1, and B4 were 6.0, 6.0, 8.4, and 10.3, respectively. There was no significant difference in pH between the A4 and B4 points near the intersection of the A line and the B line. From the analysis of Figure 9, it can be seen that on the concentric circle with the center of the regular hexagon as the center, the electric field intensity was the highest on line A between the center of the regular hexagon and the six corners (A line), and the lowest on the line (B line) between the center of the regular hexagon and the midpoint of the two adjacent corners of the hexagon. Under the action of an electric field, the $H^+$ produced by electrolysis in the anode moved to the cathode, and the $OH^-$ produced by the electrolysis in the cathode moved to the anode. The migration of $H^+$ and $OH^-$ changes the soil pH value. The soil pH near the anode area decreased, and the soil pH near the cathode area increased. Because the migration rate of $H^+$ is higher than that of $OH^-$ [37], the acidic peak appears near the A2 point in the control group. There is a lack of transition between the strongly acidic region and the strongly alkaline region, which is prone to focus effect and results in heavy metal precipitation. The B line is in the strong alkaline area, and the heavy metals precipitate, which is not conducive to the migration of heavy metals. Microbes modify the soil pH by producing various organic and inorganic

acids and other metabolites through a mechanism known as rhizosphere acidification [5]. Burkholderia cepacia is a kind of phosphate-solubilizing bacteria (PSB) [8]. Phosphate-solubilizing microbes primarily secrete oxalic, malic, fumaric, acetic, tartaric, malonic, glutanic, propoinic, butyric, lactic, gluconic, 2-keto gluconic, glyconic, and oxalic acids [9]. The BC group produced organic acids through *Burkholderia cepacia* metabolism and reduced the soil pH of the remediation unit. It was beneficial to increase the solubility of heavy metal hydroxides, carbonates, and phosphates and improved the ability of migration and transformation of heavy metals.

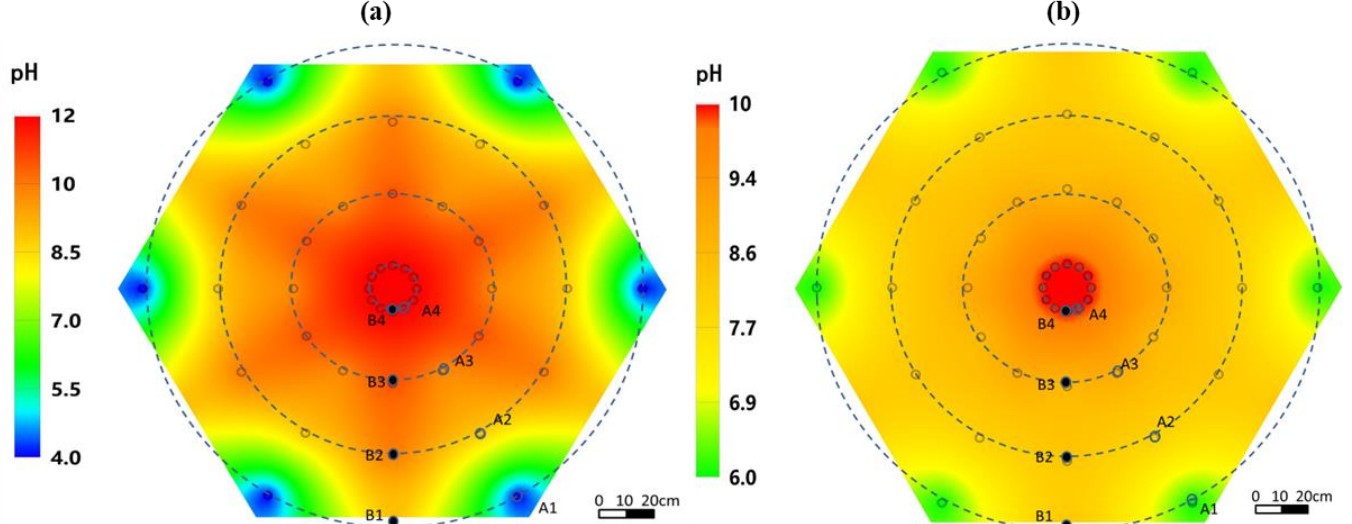

**Figure 9.** Spatial distribution of pH after EK-PRB remediation without (**a**) and with (**b**) *Burkholderia cepacia* pre-treatment.

### 3.5. Changes of Pb(II) Concentrations after Microorganism-EK-PRB Remediation

To evaluate the impact of the *Burkholderia cepacia* pre-treatment on $Pb^{2+}$ removal by EK-PRB technology, the $Pb^{2+}$ concentration in the PRB material installed at the anode and the cathode is shown in Figure 10a, and the residual $Pb^{2+}$ concentrations in anode and cathode soils are shown in Figure 10b. After EK-PRB remediation, the $Pb^{2+}$ adsorption capacity of the FeMn-ATP material in the anode and the anode was significantly higher in the BC group than in the control group ($p < 0.05$). The addition of microorganisms increased the $Pb^{2+}$ adsorption capacity of the anode and cathode soil by 69.1% and 42.1%, respectively (Figure 10a). After EK-PRB remediation, the concentration of residual $Pb^{2+}$ in the anode soil was significantly lower than that in the control group without microorganism addition ($p < 0.05$), and the concentration of residual $Pb^{2+}$ in the cathode soil was slightly lower than that in the control group without microbial addition ($p < 0.05$). The above results showed that the remediation effect of microbial-EK-PRB on lead-contaminated soil was better than that of EK-PRB without microorganisms, especially in anode soil; the removal rate of lead was increased by 26% with the addition of microorganisms.

Based on the Pb residue concentration at each point of line A and line B, the spatial distribution of the Pb residue concentration can be calculated using the interpolation method using EVS2020 software. The spatial distribution of the residual $Pb^{2+}$ concentration in the soil after remediation is shown in Figure 11 (the measured value drawn by interpolation method on EVS software). In the control group without microorganism addition, the residual concentration of lead at the B3 point was slightly lower, and the residual concentration of lead in other areas did not decrease (Figure 11a). In the BC group with *Burkholderia cepacia* pre-treatment, the lead concentration at each sampling point on line A decreased, and the lead removal rate at the six anode A1 points near the hexagonal apex was the highest, reaching 47.6%, and the lead at each point of line A and line B was removed after remediation (Figure 11b). Pb in soils mainly exists as a solid, such as Pb $(OH)_2$, $PbCO_3$,

and PbSO$_4$. The vast majority of Pb salts are poorly water-soluble, with very low levels of water-soluble Pb in soil solutions. Under alkaline conditions, Pb$^{2+}$ is more likely to form precipitates [38]. Comparing the results from the two groups, it can be seen that Pb removal after microorganism addition was higher, and acid production by *Burkholderia cepacia* decreased soil pH due to the fact that the soil pH directly controls the solubility of heavy metals.

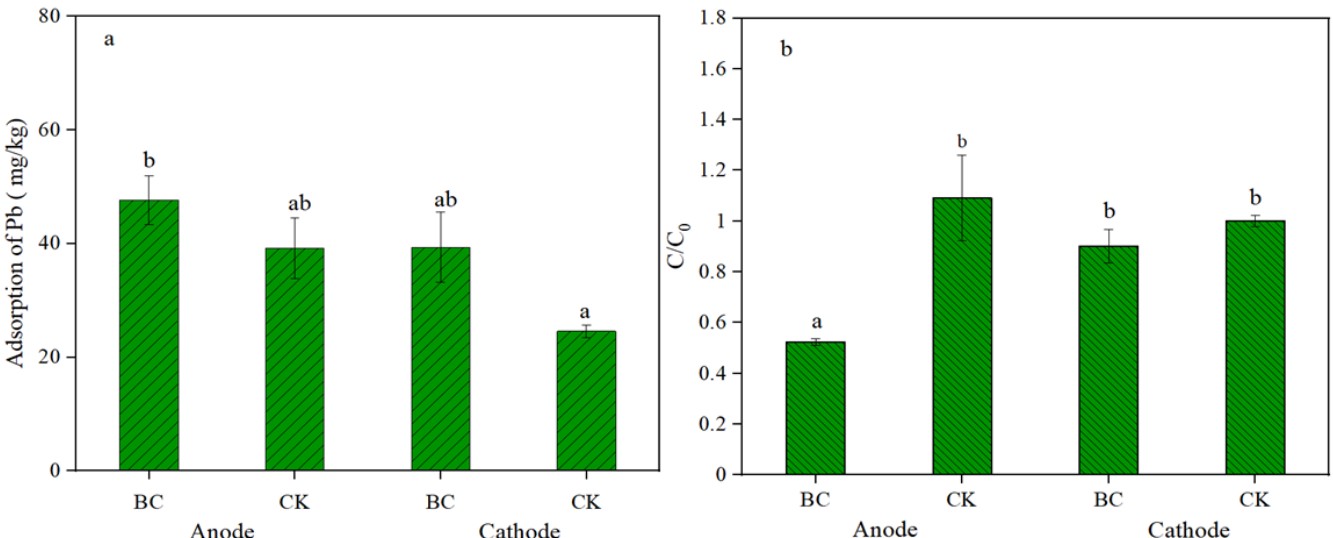

**Figure 10.** Changes in PRB sorption in *Burkholderia cepacia*-enhanced electroremediated soils (**a**). Residual concentration of Pb in soil (**b**). Note: Data in the figure indicate mean of replications $\pm$ SE. Different letters above bar columns indicate significant differences at $p < 0.05$.

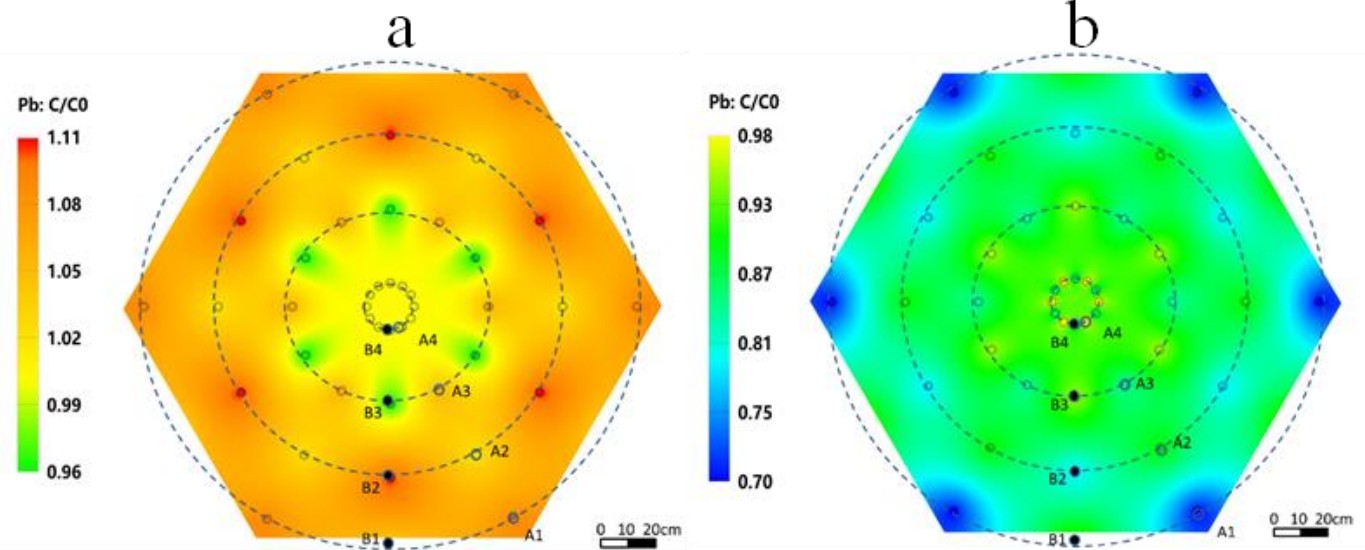

**Figure 11.** Spatial distribution of pb after EK-PRB remediation without (**a**) and with (**b**) *Burkholderia cepacia* pre-treatment.

### 3.6. Changes of Pb (II) Fractions after Microorganism-EK-PRB Remediation

In order to further study the effect of different pre-treatments on different chemical fractions of Pb$^{2+}$, a five-step continuous extraction method was used to determine the chemical forms of residual Pb$^{2+}$ in the soils at the end of the EK experiments. In order to further study Pb$^{2+}$ removal by microorganism-EK-PRB technology, a five-step continuous extraction method was used to determine the chemical forms of Pb$^{2+}$ in the soils at the end of the experiments. Figure 12 shows the morphological changes of lead in the soil

before and after microorganism-EK-PRB remediation. On line A (Figure 12a), after 14 days' remediation of A1, the ratio of the F3 decreased by 10.4%, and the ratio of the F2 and F4 fractions increased. At the A4 point, the ratio of F2 and F5 increased. The decrease in the ratio of F3 and the increase in F2 at A4 point is because the reducible lead in the soil mainly exists in iron and manganese oxide bound form and the low redox potential of cathodic soil Pb to the decomposition of iron and manganese oxide. The organic acids produced by the metabolism of *Burkholderia cepacia* microorganisms dissociate into protons and low molecular weight organic acids in the soil, resulting in a decrease in soil pH and oxidation potential, which promotes the weathering and dissolution of primary heavy metal minerals in the soil [27]. It reduces the content of reducible lead and increases the content of soluble lead [39]. On line B (Figure 12b), at B1, the ratio of F3 decreased by 44.0%, and F5 increased by 44.7%. At B4, the ratio of F5 decreased by 20.5%, and F3 at point B4 increased by 17.5%. There was a positive correlation between the soil water content and Fe-Mn oxidation state [40]. The soil moisture content at the B1 point was lower, resulting in a decreased proportion of the Fe-Mn oxidation state. The relative abundance of *Burkholderia cepacia* at the B4 point was relatively high (Supplementary material Figure S1). *Burkholderia cepacia* has a strong activity for dissolving phosphorus which is a kind of PSB [8]. There are extracellular polyphosphatase (PPX) and polyphosphate kinase (PPK) in the cell. In the acidic environment, stress-induced protein stimulates the activity of PPK, promotes the synthesis of intracellular polyphosphate polymers, and enhances the absorption of inorganic phosphate [41]. In the alkaline environment with pH 7.5, the activity of PPX was enhanced, the polymers were degraded, and inorganic phosphates were released out of the cells. *Burkholderia cepacia* has a functional phosphate dissolving gene and can secrete gluconic acid and 2-keto gluconic acid. These two organic acids chelate with heavy metals and can dissolve insoluble phosphates precipitated with heavy metals, increase the solubility of phosphorus, and convert them into soluble phosphates [42]. The low-molecular-weight organic acids solubilize the fixed inorganic P by lowering the soil pH, chelating cations, and competing with phosphate ($PO_4^-$) for adsorption sites in the soil [10]. Phosphorus dissolution promoted phosphate precipitation and lead dissolution, and the proportion of the residual state decreased. This accumulation could be attributed to the increased heavy metal mobilization due to biosurfactants produced by endophytic bacteria, as previously reported by various researchers [6]. Therefore, the mechanism of *Burkholderia cepacia* promoting the transformation of lead forms in soil was the combined action of acid production and dissolved phosphorus.

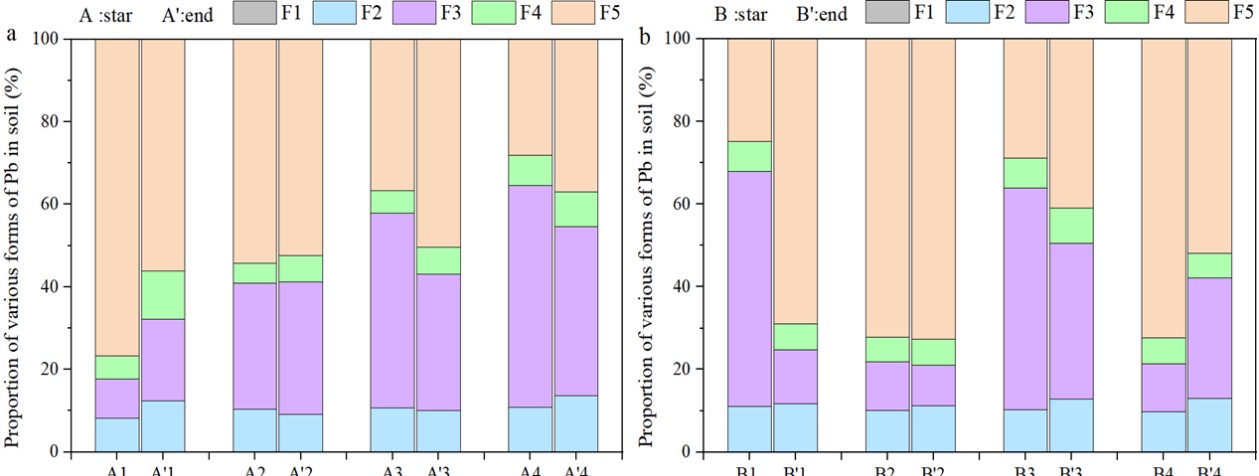

**Figure 12.** Morphological distribution of Pb at each sampling site on Line A (**a**) and Line B (**b**) (F1: exchangeable fraction, F2: carbonate fraction, F3: Fe-Mn oxidation fraction, F4: organic fraction, F5: residual fraction).

## 4. Conclusions

The following conclusions can be drawn:

The permeable reaction wall (FeMn-ATP) of iron and the manganese-modified atta-pulgite has the best adsorption effect for $Pb^{2+}$, reaching 2521 mg/kg.

The actual contaminated soil was repaired by adding *Burkholderia cepacian*-enhanced PRB (FeMn-ATP)-hexagonal rod electrode matrix; the lead adsorption capacity of the FeMn-ATP composite PRB materials was significantly higher than that of the control group without microorganisms ($p < 0.05$), especially the lead adsorption was increased by 69.1% (anode) and 42.1% (cathode), respectively. The concentration of residual lead in the anode soil was significantly lower than that in the control group ($p < 0.05$), and the removal rate of lead was increased by 26% (anode).

The mechanism of *Burkholderia cepacia* enhancing soil lead electro remediation is that acidogenesis reduces the competitive adsorption of soil pH, $H^+$ with lead ions adsorbed on the surface of soil particles such as carbonate, hydroxide, iron, and manganese oxide, and promotes the partial dissolution of soil lead precipitation. At the same time, phosphorus dissolution dissolves the precipitation of phosphorites. To sum up, *Burkholderia cepacia* can promote the transformation of $Pb^{2+}$ from stable forms to unstable forms that are easy to migrate and remove through microbial metabolic activities to enhance the effect of electrokinetic remediation. It is of great guiding significance to engineering applications.

**Supplementary Materials:** The following supporting information can be downloaded at: https://www.mdpi.com/article/10.3390/su141811440/s1, Figure S1: The main microorganisms distribution in soil of phylum level before and after the experiment (A1: Before remediation; A1': After remediation).

**Author Contributions:** Conceptualization, Y.H. and C.H.; investigation, Y.H.; resources, C.H.; data curation, F.W.; writing—original draft preparation, Y.H. and L.Y.; writing—review and editing, L.Y. and F.W.; supervision, C.H. and F.W.; project administration, C.H.; funding acquisition, C.H. All authors have read and agreed to the published version of the manuscript.

**Funding:** This study was financially supported by Shanghai Science and Technology Project (No. 19DZ1205202), Natural Science Foundation of China (No. 41971055) and State Key Laboratory of Pollution Control and Resource Reuse Foundation (No. PCRRF19003).

**Institutional Review Board Statement:** Not applicable.

**Informed Consent Statement:** Not applicable.

**Data Availability Statement:** Not applicable.

**Conflicts of Interest:** The authors declare no conflict of interest.

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
