# Peer review of "Burkholderia cepacia Enhanced Electrokinetic-Permeable Reaction Barrier for the Remediation of Lead Contaminated Soils"

_sustainability, doi:10.3390/su141811440_

Round 1

Reviewer 1 Report

Most of my concerns are positive to this manuscript. Authors nicely presented the data and i really like the idea of combining electric current with meatl tolerant microbes. 

However some key points should be taken into consideration. 

The authors were more clear towards the electrical current part but they failed to provide description of microbial utilization and its methods in the text. 

 How the Burkholderia spp was carried and applied in the study? They tried to provide information but without any adequate refernce and too naive. 

Please italicize the bacterial name throughout the experiment. 

There should be a gap statement in the last paragraph of introduction. I believe the citations can be enriched in the introduction for the bacterial part, some are provided below

Burkholderia phytofirmans PsJN and tree twigs derived biochar together retrieved Pb-induced growth, physiological and biochemical disturbances by minimizing its uptake and

Rhizosphere bacteria in plant growth promotion, biocontrol, and bioremediation of contaminated sites: a comprehensive review of effects and mechanisms

Please elaborate the mechnisms used by bacteria to remediate Pb in association with electrodes. The authors speculated that the bacteria reduced soil pH and Pb dissolution as the main mechanisms here but failed to refer and interpret with previous studies. 

Insights into the interactions among roots, rhizosphere, and rhizobacteria for improving plant growth and tolerance to abiotic stresses: a review

Remove the general statements from conclusion. rest is fine.

Author Response

Reviewer #1:

Most of my concerns are positive to this manuscript. Authors nicely presented the data and i really like the idea of combining electric current with meatl tolerant microbes.

Response: The comments from Reviewer #1 are appreciated. Please note: line numbers in the Reviewer’s comments refer to the last version of the manuscript, while the line numbers in the Authors’ answers refer to the current revised manuscript.

Comment 1:

The authors were more clear towards the electrical current part but they failed to provide description of microbial utilization and its methods in the text.

How the Burkholderia spp was carried and applied in the study? They tried to provide information but without any adequate refernce and too naive. 

Response: Thanks, the part has been added in Lines 205-208.

The experiment was divided into two groups: the control group without microorgan-ism addition and the experimental group with microorganism. After sterilizing the actual contaminated soil with a sterilizer at 121℃ for 3 consecutive days, add Burkholderia cepacia solution to each group of experimental soil to stir evenly, add an appropriate amount of deionized water which keep the soil moisture content at 30% [7]”

Comment 2:

Please italicize the bacterial name throughout the experiment.

Response: Thanks, A total of 22 points in the full text have been modified.

Comment 3:

There should be a gap statement in the last paragraph of introduction. I believe the citations can be enriched in the introduction for the bacterial part, some are provided below

Burkholderia phytofirmans PsJN and tree twigs derived biochar together retrieved Pb-induced growth, physiological and biochemical disturbances by minimizing its uptake and

Rhizosphere bacteria in plant growth promotion, biocontrol, and bioremediation of contaminated sites: a comprehensive review of effects and mechanism

Response: Thanks, the part has been added in Lines 107-113.

“Qudsia et al present the release of organic acid by microorganisms to solubilize inor-ganic P bound to soil colloids is an important mechanism in which COO (carboxyl group) and OH (hydroxyl ion) act as chelators of cations such as Fe, Al3+, and Ca2+ and compete for P adsorption sites in soil [5]. Naveed et al found the ameliorative role of applied biochar and seed inoculation with B. phytofirmans PsJN through improved growth, physiological, biochemical and anti-oxidative defence responses in mung bean grown under normal and Pb-spiked soils [6].”

Comment 4:

Please elaborate the mechnisms used by bacteria to remediate Pb in association with electrodes. The authors speculated that the bacteria reduced soil pH and Pb dissolution as the main mechanisms here but failed to refer and interpret with previous studies.

Insights into the interactions among roots, rhizosphere, and rhizobacteria for improving plant growth and tolerance to abiotic stresses: a review

Response: Thanks for this comment. The following has been added.

  1. “Microbes modify the soil pH by producing various organic and inorganic acids and other metabolites through a mechanism known as rhizosphere acidification [5].Burkholderia cepacia is a kind of phosphate-solubilizing bacteria (PSB) [8].Phosphate-solubilizing microbes primarily secrete oxalic, malic, fumaric, acetic, tar-taric, malonic, glutanic, propoinic, butyric, lactic, gluconic, 2-keto gluconic, glyconic, and oxalic acids [9].” Which is in lines 381-385.
  2. “…dissolving phosphorus which is a kind of PSB [8].” Which is in line 457.
  3. “The low-molecular-weight organic acids solubilize the fixed inorganic P by lowering the soil pH, chelating cations, and competing with phosphate (PO4) for adsorption sites in the soil [10].” Which is in lines 466-468.
  4. “This accumulation could be attributed to the increased heavy metal mobilization due to biosurfactants produced by endophytic bacteria as previously reported by various researchers [6].” Which is in lines 470-472.

Comment 5: Remove the general statements from conclusion. rest is fine.

Response: Thanks, “Compared with the hexagonal matrix EK-PRB without adding microorganisms, after” was deleted which is in line 487.

Reviewer 2 Report

Abstract:

This section presents the most important results, but it would have been useful to briefly emphasize their practical importance.

Row 16: leave a gap between value and measurement unit.

Keywords:

It is preferable that in this section words that are also found in the title are not repeated.

Introduction:

This section needs to be improved. The English language is quite deficient, therefore they must improve this aspect. Many sentences resemble to telegrams, sometimes unrelated to the previous sentence, which makes the information chaotic.

Lines 28-30: replace “It is absorbed and accumulated by human, who causes serious harm to human health [1]. Thus, it may pose a serious threat to human health directly or indirectly [2].” by “In addition, its direct or indirect absorption and subsequent accumulation in human tissues will cause serious health problems [1,2]“.

Lines 30-32: First the authors talk about human health and then they go directly to soil pollution in general (not strictly due to Pb) in the first-time mentioned regions (country should be mentioned too). In addition, the sources of pollution in the respective region should also be explained here, because the readers do not have this information and cannot get an overall picture of the respective situation.

Lines 33-34: The remediation of Pb-polluted soils can be achieved through physical and chemical technologies [4], and through bioremediation [5].

Line 40: stick to one word, use remediation not repair.

Lines 42-43: As stated in our previous study, the combination of EK and PRB possessed synergistic effect for heavy metal removal [7]”.

Line 48: “Mahabadi et al. have studied that natural zeolite reduced the leaching” – replace by “”Mahabadi et al. have shown that natural zeolite reduced the leaching”.

Line 49-50: “But it is difficult to collect from soil in practice because of 49 its loose structure.” This is where the subject is missing! Who (what) is difficult to sample? Natural zeolite or Cd? Pay attention to English grammar (and not only) and be specific!

The authors jump chaotically from one aspect to another, from natural zeolites to nanomaterials. There are so many types of nanomaterials, obtained by different methods and from different sources, that from the way you express yourself it is understood that natural zeolites are nanomaterials.

Lines 56-59: Rephrase, so that what you present is understood more clearly.

Line 57: Fe instead of iron, since you used the chemical symbol so far. Again, make connections between informations, between sentences. And use new paragraph when you switch to another subject.

Line 68: Add a sentence introducing the topic of electrodes before talking about them here.

Lines 73-74: Use the chemical symbols.

Line 74: Use new paragraph for the subject of using microorganisms. Separate the subject of electrodes from the bioremediation is logical.

Line 91: Macaskie, not macaskie.

Materials and methods

Line 113: leave a gap between values and measurement unit.

Table 1: Values, not values.

Line 118 and elsewhere: use italic for microorganism’s scientific names.

Line 145: leave a gap between values and measurement unit.

Line 152: Heavy

Line 154-155: leave a gap between values and measurement unit. Check and correct in the entire manuscript.

Line 159: Pb2+.

Place Figure 2 above its caption (name).

Lines 265-266: Check and correct. There are 4 arrows not in place. They overlap the equations.

Place Figure 4 above its caption (name).

Check all the material and correct both in the Figures caption and the text, because it is chaotically numbered. From Figure 4 you go to Figure 7, then to Figure 5 and so on!

Line 314: cm-1. Use superscript.

Figure 10 caption – cathode not cathde.

Conclusions

Line 450: delete it.

Line 466: At

Add here a concluding remark, an overall importance of your findings.

Author Response

Reviewer #2:

The comments from Reviewer #2 are appreciated. Please note: line numbers in the Reviewer’s comments refer to the last version of the manuscript, while the line numbers in the Authors’ answers refer to the current revised manuscript.

Abstract:

Comment 1:

This section presents the most important results, but it would have been useful to briefly emphasize their practical importance.

Response: Thanks for this comment. the part has been added in Lines 15-17.

“The PRB is a tough material, which can be plugged and pulled out in engineering. The heavy metals adsorbed on it can be extracted from the soil, eliminating the risk of subsequent secondary pollution. Therefore, it has a strong operational ability.”

Comment 2:

Row 16: leave a gap between value and measurement unit.

Response: Thanks, the gap between value and measurement unit has been added which is in line 18.

Keywords:

Comment 3:

It is preferable that in this section words that are also found in the title are not repeated.

Response: Thanks, “electrokinetic; lead; permeable reaction barrier” were deleted and “heavy metal; PRB” were added which is in lines 26-27.

Introduction:

Comment 4:

This section needs to be improved. The English language is quite deficient, therefore they must improve this aspect. Many sentences resemble to telegrams, sometimes unrelated to the previous sentence, which makes the information chaotic.

Response: Thanks for this comment. The whole introduction has been revised based on your valable comments.

Comment 5:

Lines 28-30: replace “It is absorbed and accumulated by human, who causes serious harm to human health [1]. Thus, it may pose a serious threat to human health directly or indirectly [2].” by “In addition, its direct or indirect absorption and subsequent accumulation in human tissues will cause serious health problems [1,2] “.

Response: Thanks, this part has been replaced which is in lines 33-35.

Comment 6:

Lines 30-32: First the authors talk about human health and then they go directly to soil pollution in general (not strictly due to Pb) in the first-time mentioned regions (country should be mentioned too). In addition, the sources of pollution in the respective region should also be explained here, because the readers do not have this information and cannot get an overall picture of the respective situation.

Response: Thanks for this comment. The lines 30-32 has been replaced by “The China Soil Pollution Survey published in 2014 indicated that 16.1 % of arable land is polluted and inorganic pollutants, particularly cadmium (Cd) is found in high quan-tities in 7 % of surveyed sites [3]” which is in lines 35-37.

Comment 7:

Lines 33-34: The remediation of Pb-polluted soils can be achieved through physical and chemical technologies [4], and through bioremediation [5].

Response: Thanks, this part has been modified which is in lines 41-43.

Comment 8:

Line 40: stick to one word, use remediation not repair.

Response: Thank, the “repair” has been replaced by remediation which is in line 48.

Comment 9:

Lines 42-43: “As stated in our previous study, the combination of EK and PRB possessed synergistic effect for heavy metal removal [7]”.

Response: Thanks, this part has been modified which is in line 50.

Comment 10:

Line 48: “Mahabadi et al. have studied that natural zeolite reduced the leaching” – replace by “”Mahabadi et al. have shown that natural zeolite reduced the leaching”.

Response: Thanks for this comment. this part has been modified which is in line 56.

Comment 11:

Line 49-50: “But it is difficult to collect from soil in practice because of 49 its loose structure.” This is where the subject is missing! Who (what) is difficult to sample? Natural zeolite or Cd? Pay attention to English grammar (and not only) and be specific!

Response: Thanks for this comment. This part has been modified which is in line 57.

“The natural zeolite is difficult to collect from soil in practice because of its loose structure.”

Comment 12:

The authors jump chaotically from one aspect to another, from natural zeolites to nanomaterials. There are so many types of nanomaterials, obtained by different methods and from different sources, that from the way you express yourself it is understood that natural zeolites are nanomaterials.

Response: Thanks, the part has been modified which is in line 54.

“The first is adsorbents, such as active aluminum, activated carbon, coal, natural zeolites, peat, etc which….”

Comment 13:

Lines 56-59: Rephrase, so that what you present is understood more clearly.

Line 57: Fe instead of iron, since you used the chemical symbol so far. Again, make connections between informations, between sentences. And use new paragraph when you switch to another subject.

Response: Thanks, this part has been modified which is in lines 66-69.

“The zero-valent Fe / Cu nanoparticles in contaminated soil reduction efficiency of Cr (VI c was more than 99% in weakly acidic conditions, and the price of this kind of PRB is high which requires the morphology of contaminated soil and the valence of heavy metals which its practical application is limited [4].”

Comment 14:

Line 68: Add a sentence introducing the topic of electrodes before talking about them here.

Response: Thanks, the sentence introducing the topic of electrodes has been added in lines 77-81.

“Electrokinetic remediation refers to the application of direct current at both ends of the contaminated soil to form a voltage gradient, and the pollutants in the soil are transferred to both ends of the electrode by electrodialysis, electromigration and elec-trophoresis under the action of electric field, so as to achieve the technology of remediation contaminated soil.”

Comment 15:

Lines 73-74: Use the chemical symbols.

Response: Thanks, the “cadmium” and “lead” have been changed to “Cd” and “Pb” which is in line 87.

Comment 16:

Line 74: Use new paragraph for the subject of using microorganisms. Separate the subject of electrodes from the bioremediation is logical.

Response: Thanks for this comment. The introduction of microorganisms has been divided into separate paragraph which is in line 88.

Comment 17:

Line 91: Macaskie, not macaskie.

Response: Thanks, the word has been revised which is in line 104.

Materials and methods

Comment 18:

Line 113: leave a gap between values and measurement unit.

Response: Thanks, the gap between “50” and cm” has been added in line 132.

Comment 19:

Table 1: Values, not values.

Response: Thanks, the word has been revised which is in line 136.

Comment 20:

Line 118 and elsewhere: use italic for microorganism’s scientific names.

Response: Thanks, the total of 22 points in the full text have been modified.

Comment 21:

Line 145: leave a gap between values and measurement unit.

Response: Thanks, three gap has been added which is in line 164.

Comment 22:

Line 152: Heavy

Response: Thanks, the word has been revised which is in line 171.

Comment 23:

Line 154-155: leave a gap between values and measurement unit. Check and correct in the entire manuscript.

Response: Thanks, the eighteen gaps has been added in lines 174-182. Eight gaps have been added in lines 196, 200, 229,230, 232, 238, 266, 267, respectively.

Comment 24:

Line 159: Pb2+.

Response: Thanks, the “Pb2+” has been changed to “Pb2+” which is in line 179.

Comment 25:

Place Figure 2 above its caption (name.

Response: Thanks, Figure 2 has been placed above its caption (name) which is in line 213.

Comment 26:

Lines 265-266: Check and correct. There are 4 arrows not in place. They overlap the equations.

Response: Thanks, this part has been revised which is in lines 259-260, 286-287, respectively.

Comment 27:

Place Figure 4 above its caption (name).

Check all the material and correct both in the Figures caption and the text, because it is chaotically numbered. From Figure 4 you go to Figure 7, then to Figure 5 and so on!

Response: Thanks, Figure 4 has been placed above its caption (name) which is in line 288.

 “Figure 7”, “Figure 5” and “Figure 6” have changed “Figure 5”, “Figure 6” and “Figure 7”, respectively.

Comment 28:

Line 314: cm-1. Use superscript.

Response: Thanks, the “cm-1” has been changed to “cm-1” which is in line 336.

Comment 29:

Figure 10 caption – cathode not cathde.

Response: Thanks, the part has been revised which is in line 408.

Conclusions

Comment 30:

Line 450: delete it.

Response: Thanks, “The adsorption kinetics accords with the quasi-first-order kinetic equation which is mainly physical adsorption” has been deleted which is in lines 485-486.

Comment 31:

Line 466: At

Response: Thanks, the “at” has been changed to “At” which is in line 497.

Comment 32:

Line 466: Add here a concluding remark, an overall importance of your findings.

Response: Thanks for this comment.

“To sum up, Burkholderia cepacia can promote the transformation of Pb2+ from stable forms to unstable forms that are easy to migrate and remove through microbial metabolic activities to enhance the effect of electrokinetic remediation. And it is of great guiding significance to the engineering application.” which is in lines 498-502.

  1. Pierart, A.; Shahid, M.; Sejalon-Delmas, N.; Dumat, C. Antimony bioavailability: knowledge and research perspectives for sustainable agricultures. J Hazard Mater 2015, 289, 219-234, doi:10.1016/j.jhazmat.2015.02.011.
  2. Zwolak, A.; Sarzyńska, M.; Szpyrka, E.; Stawarczyk, K. Sources of Soil Pollution by Heavy Metals and Their Accumulation in Vegetables: a Review. Water, Air, & Soil Pollution 2019, 230, doi:10.1007/s11270-019-4221-y.
  3. 3Yin Ping, L.L., Chen Bin, Xiao Guoqiang, Cao Ke, Yang Jilong, Li Meina, Duan Xiaoyong, Qiu Jiandong, Hu Yunzhuang, Wang Lei, Sun Xiaoming. Coastal zone geo-resources and geo-environment in China. Geology in China 2017, 44, 842-854, doi:10.12029/gc20170502.
  4. Hu, B.; Song, Y.; Wu, S.; Zhu, Y.; Sheng, G. Slow released nutrient-immobilized biochar: A novel permeable reactive barrier filler for Cr(VI) removal. Journal of Molecular Liquids 2019, 286, doi:10.1016/j.molliq.2019.04.153.
  5. Ojuederie, O.B.; Babalola, O.O. Microbial and Plant-Assisted Bioremediation of Heavy Metal Polluted Environments: A Review. Int J Environ Res Public Health 2017, 14, doi:10.3390/ijerph14121504.
  6. Wan, J.; Li, Z.; Lu, X.; Yuan, S. Remediation of a hexachlorobenzene-contaminated soil by surfactant-enhanced electrokinetics coupled with microscale Pd/Fe PRB. J Hazard Mater 2010, 184, 184-190, doi:10.1016/j.jhazmat.2010.08.022.
  7. He, C.; Hu, A.; Wang, F.; Zhang, P.; Zhao, Z.; Zhao, Y.; Liu, X. Effective remediation of cadmium and zinc co-contaminated soil by electrokinetic-permeable reactive barrier with a pretreatment of complexing agent and microorganism. Chemical Engineering Journal 2021, 407, doi:10.1016/j.cej.2020.126923.
  8. Yu, X.; Muhammad, F.; Yan, Y.; Yu, L.; Li, H.; Huang, X.; Jiao, B.; Lu, N.; Li, D. Effect of chemical additives on electrokinetic remediation of Cr-contaminated soil coupled with a permeable reactive barrier. R Soc Open Sci 2019, 6, 182138, doi:10.1098/rsos.182138.
  9. Mahabadi, A.A.; Hajabbasi, M.A.; Khademi, H.; Kazemian, H. Soil cadmium stabilization using an Iranian natural zeolite. Geoderma 2007, 137, 388-393, doi:10.1016/j.geoderma.2006.08.032.
  10. Wang, M.; Chen, S.-B.; Li, N.; Ma, Y.-B. A review on the development and application of nano-scale amendment in remediating polluted soils and waters. Chinese Journal of Eco-Agriculture 2010, 18, 434-439, doi:10.3724/sp.J.1011.2010.00434.

Round 2

Reviewer 1 Report

I am satisfied with the revision. The authors have carried out revision.

Reviewer 2 Report

In the revised version, the authors have responded to all the comments and requests for necessary corrections in the article. I believe that this revised version corresponds to the level of a scientific research article and I recommend its publication. I congratulate the authors and wish them success in their work.